

# Simulation of future impact of black carbon emissions from the Northern and Transpolar Sea routes on Arctic sea ice

Anna Poltronieri[1], Nils Bochow[1], and Martin Rypdal[1]

[1]Department of Mathematics and Statistics, Faculty of Science and Technology, UiT - The Arctic University of Norway, Norway.

**Correspondence:** Anna Poltronieri (anna.poltronieri@uit.no)

**Abstract.** As sea ice decreases, navigation in the Arctic is becoming more feasible, and new routes are likely to emerge. However, the impact of these potential routes on sea ice remains uncertain. In this study, we compare the regional impacts of two major Arctic routes: the Northern Sea Route (NSR) and the Transpolar Sea Route (TSR). Using the Community Earth System Model (CESM2), we simulate black carbon (BC) emissions until 2050 along these routes and assess their effects on
Arctic sea ice (ASI). We focus on regional changes in net shortwave (SW) radiation, sea ice extent, and surface temperature. While our study does not account for other pollutants that could counteract BC effects, our results reveal significant differences in ASI's response between routes. The TSR, in particular, exerts a stronger and more widespread influence on ASI than the NSR across all seasons, especially in increasing net SW radiation over the ice.

## 1   Introduction

Ocean shipping is a fundamental component of global trade, with around 80% of goods transported by sea (Statista Research Department, 2024). Since 1990, the volume of seaborne trade has more than doubled, rising from 400 Mt to nearly 1,100 Mt by 2021 (Statista Research Department, 2024). This global increase in maritime activity is also reflected in the Arctic, where the number of unique ships entering the Arctic Polar Code area has increased by 37% between 2013 and 2023 (PAME, 2024). Navigation in this region has become increasingly feasible due to the rapid decline of the Arctic sea ice (ASI), attracting

interest in establishing new shipping routes (Lasserre, 2014; Stocker et al., 2020). Numerous studies predict that the Arctic could become ice-free during summer (area below 1 million $\text{km}^2$) before the middle of this century, opening the region for maritime activity (Kim et al., 2023; Jahn et al., 2024; Poltronieri et al., 2024a; Notz and SIMIP Community, 2020). In the future, the Northwest Passage, the Northern Sea Route (NSR), and the Transpolar Sea Route (TSR) may potentially replace the traditional pathways through the Panama and Suez Canals (Mudryk et al., 2021). The NSR lies in the Kara, Laptev, East

Siberian, and Chukchi seas, connecting Western Eurasia to the Asia-Pacific basin (Chaudhury, 2024). The TSR crosses the center of the Arctic, connecting the Atlantic and Pacific oceans (Buixadé Farré et al., 2014). Contrary to the NSR, the TSR is not a coastal route and lies in international high waters, notably reducing the travel distance between Europe and Asia (Østreng et al., 2013).





While Arctic navigation offers shorter transit times, reduced fuel consumption, and lower overall costs (Theocharis et al.,
2018), it also raises several safety and environmental concerns (Svavarsson et al., 2021; Sweeney et al., 2022). For example, the
year-to-year variability in the summer ASI cover is expected to increase in the future, making forecasts of ice-free conditions
unreliable and potentially unsafe for navigation (Poltronieri et al., 2024b). At the same time, the expansion of Arctic shipping
is not limited to summer. Between 2013 and 2022, winter sailing has increased significantly, leading to ships operating in
hazardous sea ice conditions three times more often than before (Müller et al., 2023). The Arctic ecosystems and water bodies
are directly threatened by noise pollution, fossil fuel spills, and the introduction of non-indigenous species (Qi et al., 2024).
The compounds released in the atmosphere from burning fossil fuels increase air pollution and disrupt radiative forcing (Qi
et al., 2024). In particular, $SO_2$ emissions contribute to the formation of highly reflective clouds, which create a cooling effect
that could reduce the temperature at the surface (TAS) in the Arctic by 1°C by the end of the century (Stephenson et al., 2018).
On the other hand, black carbon (BC) deposits contribute to Arctic warming by darkening the ice and snow surfaces, reducing
their albedo and increasing heat absorption (Corbett et al., 2010; Ødemark et al., 2012). This effect introduces significant
uncertainties in estimating global radiative forcing, particularly in the Arctic during spring (Pedersen et al., 2015). The high
albedo of the sea ice and snow plays a crucial role in regulating the temperature in the Arctic, as these surfaces typically reflect
more than half of the incoming shortwave (SW) radiation (Shokr and Sinha, 2023). As the sea ice melts, more open water is
exposed, decreasing the reflectivity of the Arctic surface. In the last decades, the loss of sea ice has led to an effective reduction
in albedo from 0.52 to 0.48 and an increase in absorbed SW of around 6 W/m$^2$ since 1979 over the Arctic region (Pistone
et al., 2014). Generally, more absorbed SW radiation leads to higher TAS and consequently to even further reduction in sea ice
cover (Letterly et al., 2018).

The spatial origin of BC emissions is particularly important for the Arctic. In contrast to the BC emitted at lower latitudes,
which is subsequently transported to the Arctic, the BC released within the Arctic region is directly deposited on the ice and
snow, amplifying its warming effect (Stephenson et al., 2018). This localized warming effect can be up to five times stronger
compared to the TAS increase caused by the same amount of BC emitted at mid-latitudes (Sand et al., 2013). Given the
importance of regional BC emissions, it is essential to assess how different shipping routes affect the sea ice surface. In this
study, we employ the state-of-the-art, fully coupled global Earth system model CESM2 to investigate regional changes in the
Arctic due to BC deposition from shipping along two routes: the NSR and the TSR. Our findings reveal significant differences
in the Arctic's regional response to BC deposition from these routes, resulting in distinct localized impacts across all seasons.

## 2 Data and Methods

In our study, we use version 2.1.3 of the Community Earth System Model (CESM2), developed by the National Center for
Atmospheric Research (NCAR), to simulate navigation along the NSR and TSR (Danabasoglu et al., 2020). We estimate the
opening times for these routes, that is, the timing of BC injection, using the three CESM2 runs under the Shared Socioeconomic
Pathway scenario with very high greenhouse gas emissions (SSP5-8.5) provided in the latest phase of the Coupled Model
Intercomparison Project (CMIP6) (Eyring et al., 2016), as described below.





## 2.1 Routes

We define the boundaries of the NSR route as described by the NSR administration of Russia at http://www.nsra.ru/en/ofitsialnaya_informatsiya/granici_smp.html. However, because of the high variability of the ASI conditions, the boundaries
of the TSR area are not as clearly defined and this route will likely consist of several maritime paths across the central Arctic (Smith and Stephenson, 2013). Figure 1a illustrates the areas of the routes used in our experiments.

### 2.1.1 Navigability

We employ CESM2 runs from CMIP6 under the SSP5-8.5 scenario to define the moment when the TSR starts to be navigable. This scenario is chosen as it is the only one that ensures sufficient ASI loss in the central regions of the Arctic to establish
navigation along the TSR before 2050. Since the NSR is expected to open earlier than the TSR, we use these projected opening dates to determine the start of navigation along both routes (Dams et al., 2020). We collect and average the three available runs from CMIP6, and analyze the ASI concentration along the TSR path. We consider the TSR to be viable in a given month when an ice-free passage - where the sea ice concentration (SIC) is below 15% throughout the entire month - exists for at least five continuous years. The opening dates for the TSR obtained from the averaged CMIP6 runs are July 2083, August 2055,
September 2034, October 2060, November 2080, and December 2091. The dates derived from this method are in accordance with previous estimates for the opening of the TSR, which is expected to be seasonally navigable by the middle of the century (Bennett et al., 2020). According to this methodology, the TSR is not viable during the first six months of the year before 2100.

## 2.2 Shipping emissions

We estimate the future BC shipping emissions starting from a baseline observational time series ranging from 2013 to 2016,
which we average over the years to obtain a representative 12-month time series (meanBC). The emissions for 2013 are published in Figenschau and Lu (2022). The data from 2014 to 2016 are provided by the Protection of the Arctic Marine Environment's Arctic Ship Traffic Database (PAME, 2021) and are processed as described in Figenschau and Lu (2022).

We project the average time series meanBC into the future, incorporating the ASI data from the CMIP6 runs. This allows us to include both the ASI decline and variations in the seasonal cycle when extending the emissions time series into the future.
We define the extended BC emissions time series $\text{emis}_{\text{ice}}(y, m)$ for a given year $y$ and month $m$ as:

$$\text{emis}_{\text{ice}}(y, m) = \text{meanBC}(m) - k \sum_{i=1}^{y-1} \Delta \text{ice}(i, m), \tag{1}$$

with the conversion factor $k = 1 [\text{T} \cdot \text{yr/km}^2]$ and $\Delta \text{ice}(i, m)$ as the the slope of the monthly ASI time series obtained from the average of the CMIP6 models over 5-year-long intervals. Finally, we adjust the extended BC time series by adding 2.5% of the cumulative emissions from the previous years uniformly over the months to account for economic growth:

$$\text{BC}(y, m) = \text{emis}_{\text{ice}}(y, m) + \frac{2.5}{100} \sum_{i=1}^{y-1} \frac{1}{12} \sum_{j=1}^{12} \text{emis}_{\text{ice}}(i, j). \tag{2}$$



We obtain annual emissions with a magnitude consistent with the future projections in Chen et al. (2024) for navigation along the NSR (Fig. 1c). Their estimates are on the order of tens to hundreds of tons per year, depending on the navigational scenario.

## 2.3 Model setup

90 CESM2 is a fully coupled global Earth system model consisting of atmosphere (Community Atmosphere Model, CAM), land (Community Land Model, CLM4.0), land ice (Community Ice Sheet Model, CISM), ocean (Parallel Ocean Program version 2, POP2), river runoff (Model for Scale Adaptive River Transport, MOSART1.0), sea ice (Community Ice CodE, CICE Version 5), and wave (WaveWatch III, WW3) components (Danabasoglu et al., 2020). We perform a simulation from the year 2015 through the end of 2030 with a horizontal resolution of 0.9° latitude by 1.25° longitude for the atmosphere and land components 95 (f09) and 1° latitude by 1° longitude for the ocean and sea ice components (g17) under the SSP5-8.5 scenario. From there, we branch out six different simulations (three for the NSR scenario and three for the TSR scenario, each with slightly perturbed initial conditions) and add the BC emissions. After one spin-up year to allow for atmospheric equilibration following aerosol injection (Simmonds, 1985), we run the model until the end of 2050. The BC emissions are spread uniformly over the NSR and the TSR areas at the bottom layer of the atmosphere. We adhere to the schedule outlined at the end of the Navigability 100 subsection to add the estimated BC emissions to the CESM2 input files for both navigational scenarios. However, because the file structure only permits emission injections every ten years and the model interpolates for the missing ones, the actual injection dates in our experiments are consequently adjusted (Fig. 1c).

## 2.4 CMIP6 data

The NSR and TSR scenarios are compared to a control ensemble consisting of four runs under the SSP5-8.5 scenario, one 105 from our model setup and the three available CESM2 runs in CMIP6. The variable for the extent of the snow cover over sea ice is not available for CESM2 in CMIP6. Therefore, we analyze the snow cover thickness. The variables we consider in our analysis, as defined in CMIP6, are: `rss` (Net Shortwave Surface Radiation), `siconc` (Sea Ice Area Percentage), `clt` (Total Cloud Cover Percentage), `tas` (Near-Surface Air Temperature), and `sisnthick` (Snow Thickness). We also utilize `areacello` (Grid Cell Area for Ocean Variables) and `areacella` (Grid Cell Area for Atmospheric Grid Variables) for 110 the spatial averages. Since no melt pond-related variables are available in CMIP6 and we only perform one control run with our model setup, we compare the fraction of radiation-effective melt ponds - ponds not covered by ice that can thus affect the surface albedo (Diamond et al., 2021) - between the NSR and TSR scenarios directly. This approach allows us to at least exclude the contribution of melt ponds from the differences in net SW radiation observed between the two scenarios.





## 2.5 Regridding

The analyses that involve comparisons of the SIC between single grid cells are performed after regridding to a 325x325 Lambert-Azimuthal equal-area grid using nearest neighbor interpolation with Climate Data Operators (CDO) (Schulzweida, 2023). All the other computations are performed on the original grids (atmospheric or oceanic).

## 3  Results

In the following, unless stated differently, we report the observed differences between each navigational scenario and the control
ensemble. The variables are calculated as the seasonal mean, averaged over the last 10 years of the experiments (2041-2050), for latitudes above 65°N. Following the methodology in Stephenson et al. (2018), we test the period-averaged differences between each navigational scenario and the control ensemble using a two-tailed Student's $t$ test. If not otherwise specified, the reported differences are statistically significant ($p < 0.05$).

In the CMIP6 ensemble, the net SW surface radiation is defined as the difference between the incoming and the outgoing
SW radiative flux at the surface (Luhar et al., 2022). Based on the seasonal partition used in our analysis - spring (March, April, and May), summer (June, July, and August), fall (September, October, and November), and winter (December, January, and February) - we present results only for spring and summer, as fall and winter only contribute minimally to the net SW radiation in the Arctic. For the sea ice extent and the TAS, we also report the most relevant results from fall and winter.

In spring, no significant change in net SW radiation at the surface under the NSR scenario is observed (Fig. 2a, Tab. A1).
However, the East Siberian, Laptev, and Barents seas show increasing trends under the TSR scenario (Fig. 2b, Tab. A2). The East Siberian and Laptev seas show modest (under 1%) and statistically insignificant sea ice extent decline under TSR scenario compared to the control ensemble (Fig. 3b, Tab. A4). Moreover, these two regions are entirely composed of grid cells with high SIC ($> 70\%$), indicating that the increased SW radiation occurs over the sea ice (Fig. 4a). Since no significant changes are observed in cloud cover (Tab. A6), snow cover over the sea ice (Tab. A8), or melt ponds (Tab. A11), this suggests that the
increased SW radiation is most likely due to a decrease in albedo caused by BC deposition. We see no significant differences in the TAS neither over the East Siberian nor the Laptev seas (Tab. A10).

The sea ice behavior in the Barents Sea is more complex. The sea ice composition differs between the TSR and control ensembles, with a decline in cells with high and medium SIC (40-70%) and an increase in cells with low SIC (15-40%) and open water (SIC $< 15\%$). Although the sea ice loss in this region is not statistically significant due to the high inter-run
variability, it is still notable, with a decline of 14.75% compared to the control ensemble. The sea ice loss in the Barents Sea under the TSR scenario is proportionally the most extensive in the entire Arctic region compared to the control ensemble, and is nearly three times greater than under the NSR scenario (Tabs. A3,A4). To analyze the change in net SW radiation over sea ice, we compare the average net SW radiation between the TSR and control scenarios over grid cells with similar SIC. Under the TSR scenario, the composition of the sea ice in the Barents Sea is evenly distributed among grid cells with low, medium,
and high SIC. The grid cells with low and high SIC show a decline in net SW radiation under the TSR scenario compared to the control scenario, while the grid cells with medium SIC show the opposite trend (Fig. 4a). Given that the extensive sea





ice loss, combined with the shift toward a more sparse ice cover, is not accompanied by a clear increase in net SW radiation over the ice surface, we cannot conclude that the observed changes in the Barents Sea are due to BC deposition on the ice. We also investigate changes in net SW radiation during the first ten years of our experiments, specifically the decade preceding the period considered in our analysis. While not statistically significant, we observe an increase in the net SW radiation under the TSR scenario in the Barents Sea during spring compared to the control ensemble (+0.98 W/m$^2$), the greatest change observed across the entire Arctic region in this period. Although this increase might potentially be the reason behind the enhanced sea ice loss in the Barents Sea, we cannot exclude that factors not considered in our analysis could be the main drivers behind the sea ice decline. Finally, we observe no significant changes in the TAS during spring under TSR (Tab. A10).

During the summer, we observe several differences in the net SW radiation between the NSR and TSR scenario. The Beaufort, Chukchi, and East Siberian seas show increased net SW radiation under the TSR scenario (Fig. 2d, Tab. A2). Although the sea ice loss is statistically significant only in the Beaufort Sea, both the Chukchi and the East Siberian seas exhibit a sea ice decline approximately four and six times greater, respectively, under the TSR scenario than under the NSR scenario when compared to the control ensemble (Fig. 3, Tabs. A3,A4). We observe no statistically significant changes in snow cover between the TSR and the control ensemble (Tab. A8). In the Chukchi Sea, we see a small, yet statistically significant, decrease in cloud cover (-0.71%). Additionally, we observe a greater fraction of melt ponds on the sea ice in the Beaufort (+2.42%) and the Chukchi (+1.79%) seas under the NSR scenario compared to TSR scenario. However, the net SW is greater under the TSR scenario, excluding the melt ponds contribution from the observed differences. In all these areas - Beaufort, Chukchi, and East Siberian seas - we observe a shift in surface composition under the TSR scenario compared to the control ensemble (Fig. 4c). The number of grid cells with medium SIC decreases, while the number of grid cells with low SIC increases. The Beaufort Sea also shows a marked increase in open water area, while the Chukchi and East Siberian seas experience a more modest increase. The comparison of net SW radiation over grid cells with similar SIC reveals a very clear increase over the entire ice-covered surface of both the Chukchi and East Siberian seas under the TSR scenario compared to the control scenario, indicating a contribution from the BC deposition (Fig. 4c). For the Beaufort Sea, the results are less clear. Net SW radiation increases over the grid cells with medium SIC, which, however, constitute only a small fraction of the total surface. In contrast, net SW radiation is slightly reduced over low SIC cells (Fig. 4c). The changes in net SW radiation over the ice surface are unclear, making it difficult to conclude with certainty whether the observed differences in the Beaufort Sea under the TSR scenario, compared to the control ensemble, are due to BC deposition. We analyze the net SW radiation in the decade prior to our analysis under the TSR scenario, observing an increase in both spring (+0.51 W/m$^2$) and summer (+1.43 W/m$^2$). While these increases might explain the strong sea ice decline in the Beaufort Sea in the following decade, it is also possible that factors aside from BC deposition are contributing to the observed sea ice loss in this region. Lastly, we note an increase by 0.46°C in the TAS.

Interestingly, several regions display comparable sea ice loss under both the NSR and TSR scenarios, significantly greater compared to the control ensemble. However, the net SW radiation exhibits notable differences between the two routes. In particular, the Laptev Sea and the Central Arctic show similar behaviors. In both regions, a comparable amount of sea ice is lost under the NSR and the TSR scenarios, respectively approximately $38 \times 10^3$ km$^2$ and $125 \times 10^3$ km$^2$. Although not statistically significant, the snow cover is thicker and the fraction of melt ponds is smaller under the NSR than under the TSR





scenario in both regions (Tabs. A7,A8&A11). In the Laptev Sea, the average SIC is slightly higher under the NSR than under the TSR scenario. In the Central Arctic, the composition is similar between the two scenarios. Lastly, the cloud cover shows no significant difference between the NSR and the TSR scenario (Tabs. A5,A6). All these factors point against increasing net

SW radiation under the NSR scenario. However, in both regions, the difference is higher under the NSR than under the TSR scenario compared to the control ensemble. In the Laptev Sea, the increase is almost two times greater under the NSR scenario (+4.67 W/m$^2$) than under the TSR scenario (+2.82 W/m$^2$). In the Central Arctic, it amounts to +3.7 W/m$^2$ under the NSR scenario and +2.64 W/m$^2$ under the TSR scenario. The change in net SW radiation, together with the similarity of the sea ice extent and composition, cloud cover, snow cover, and melt pond fraction, suggests that the reason behind the observed increase

in net SW radiation might be a drop in albedo caused by BC deposition. The grid cell analysis confirms that the increase in net SW radiation over both the Laptev Sea and the Central Arctic under the NSR scenario compared to the control ensemble happens over the ice surface (Fig. 4b). Finally, the Laptev Sea experiences a similar increase in the TAS during summer both under the NSR (+0.29°C) and under the TSR scenario (+0.33°C), although only the latter is statistically significant. The Canadian Archipelago also shows comparable sea ice loss under both the NSR and the TSR scenarios ($\sim 33 \times 10^3$ km$^2$).

However, we only find a statistically significant increase in net SW radiation under the TSR scenario (+4.61 W/m$^2$), with no significant differences neither in cloud nor in snow cover from the NSR scenario (Tabs. A5-A8). We analyze the change in net SW radiation over the ice surface and we observe a clear increase over the grid cells with medium and high SIC, while we see a decrease over the grid cells with low SIC. However, the low SIC grid cells constitute only a minor fraction of the sea ice cover in the Canadian Archipelago under the TSR scenario, confirming that the change in net SW radiation in this region is observed

over the sea ice (Fig. 4c). Lastly, we find an increase in the TAS in summer in the Canadian Archipelago under both the NSR (+0.41°C) and the TSR scenarios (+0.65°C).

The extensive sea ice loss observed in the Beaufort Sea during the summer under the TSR scenario continues in fall (-45.2$\times 10^3$ km$^2$) and winter (-50.1$\times 10^3$ km$^2$). The Barents Sea undergoes a noticeable sea ice loss during winter under the TSR scenario. Specifically, the Barents Sea shows a sea ice loss more than three times greater than under the NSR scenario

compared to the control ensemble (-54.59$\times 10^3$ km$^2$, not statistically significant). The Canadian Archipelago and Central Arctic show continued sea ice decline from summer into fall. While fall sea ice loss in the Canadian Archipelago is significant in both scenarios, confirming the trend observed in summer, in the Central Arctic it is significant only under the TSR scenario, where the sea ice cover is nearly 40$\times 10^3$ km$^2$ smaller than under the NSR scenario. The Laptev Sea also shows a statistically significant sea ice loss in fall, with a loss of 39.14$\times 10^3$ km$^2$ of sea ice cover compared to the control ensemble. Finally, we

observe strong warming in winter under the TSR scenario in both the Beaufort (+1.28°C) and the Chukchi (+1.36°C) seas, although not statistically significant.

Some regions show an unexpected response, with greater sea ice cover under our navigational scenarios compared to the control ensemble. The Baffin Bay exhibits increasing sea ice cover under the TSR scenario across all seasons (statistically significant only in spring and winter), while under the NSR scenario only in spring (not statistically significant). The Greenland

Sea also shows positive trends across all seasons, except fall, under the TSR scenario, but never under the NSR scenario. Despite the non-significance of several of these results, it is noteworthy that there are stark differences in the TAS and sea ice cover





between the NSR and the TSR scenarios during fall and winter as well (Fig. 3, Tabs. A3,A4,A9&A10). These trends may indicate emerging patterns that could potentially become significant in the context of longer simulations.

## 4 Discussion

We perform three CESM2 runs simulating BC emissions stemming from navigation along the NSR, and three along the TSR, until the year 2050. The BC is injected uniformly in the bottom layer of the atmosphere in the areas covered by the two routes. We assume that the emission levels are the same for both scenarios, to ensure comparability. Due to the high computational cost of running CESM2, we simulate only one control run with our model setup. Therefore, we additionally use three runs from the CMIP6 ensemble to build a four-run control ensemble. The underlying emission scenario for all the experiments is

the SSP5-8.5 to ensure enough sea ice loss to allow navigation along the TSR.

The goal of this study is to analyze regional changes in the Arctic due to BC deposition from the NSR and the TSR. In particular, we focus on differences in sea ice reflectivity, extent, and composition during the last 10 years of experiments (2041-2050). Additionally, we analyze changes in the TAS. Since the sea ice albedo variable is not available in the CMIP6 ensemble for CESM2 under the SSP5-8.5 scenario at the time of writing, we instead analyze the net SW radiation at the

surface. As we want to isolate the contribution of BC from the other pollutants, our experiments are idealized, and they only simulate simplified navigation scenarios. For instance, our model setup excludes contributions from added sulfates, which have been shown to lead to a cooling of the Arctic region (Stephenson et al., 2018). Therefore, in our experiments, the warming effect of the BC and the amount of downwelling SW radiation reaching the surface are not mitigated by the other shipping emissions. Moreover, it is likely that navigation along the NSR and TSR will exhibit different seasonality and amount of traffic

in the future. Our choice to keep the same emission levels and timings along both routes is not a realistic simulation of future navigation in the Arctic, but it is necessary to maintain comparability in our analysis.

Using the MASIE partition (U.S. National Ice Center et al., 2010) (Fig. 1b), we perform a regional analysis to determine if the BC deposition patterns from the two routes result in distinct regional outcomes. While we are interested in differences in net SW radiation at the surface, sea ice cover, and TAS, we also compare the cloud cover, the snow cover over the sea

ice, and the fraction of melt ponds. This is because clouds, snow, and melt ponds substantially influence the amount of SW radiation that reaches the surface and is either absorbed or reflected by the sea ice (Curry et al., 1996; Perovich et al., 2002; Perovich and Polashenski, 2012). Thus, we account for changes in these variables in our analysis, to exclude their contribution from the measured differences in net SW radiation. However, we do not consider changes in ocean circulation patterns or water temperature, which strongly influence sea ice loss and formation (Polyakov et al., 2017; Ivanov, 2023). In our study,

several regions display a significant increase in net SW radiation, often accompanied by declining sea ice cover and changes in its composition. Therefore, we additionally conduct a grid cell analysis over the sea ice surface to determine whether the increased net SW radiation is occurring over the ice — and is thus a result of BC deposition — or if it is purely due to decreased albedo following exposure of open water after sea ice loss - which could potentially be triggered by factors not considered in our analysis.





The two routes exhibit significant differences, especially in terms of absorbed SW radiation over the sea ice during spring and summer (see Results section for details and Conclusion section for a concise summary). However, we also observe some trends in fall and winter, especially under the TSR scenario. A notable result is the enhanced sea ice reduction under the TSR scenario during fall in the Central Arctic, despite the nearly identical total sea ice loss in summer across both scenarios. An important difference between the two scenarios is the spatial distribution of the sea ice loss in this region (Fig. 3c,d). Under the

NSR scenario, the loss is more pronounced near the Canadian Archipelago and along the northern coast of Greenland, while in the TSR scenario, it is primarily concentrated in the Pacific sector of the Arctic. From spring to fall, warmer, fresher water from the Pacific enters the Arctic through the Bering Strait, initiating sea ice melting (Woodgate and Peralta-Ferriz, 2021). As the sea ice cover in the Pacific sector of the Arctic is already reduced under the TSR scenario, the impact of the Pacific water entering the Arctic is likely to lead to greater sea ice loss in fall under the TSR than under the NSR scenario.

Interestingly, we also observe sea ice expansion in two regions: the Baffin Bay and the Greenland Sea. Sea ice growth along the edge of the Greenland Sea in response to Arctic shipping has been observed before (Stephenson et al., 2018). However, it has been shown to happen in response to cooling following increased sulfate emissions, which we do not include in our experiments. The Greenland Sea and the Baffin Bay are the only areas that present a consistently thicker snow cover in spring and summer under both the NSR and the TSR scenarios compared to the control ensemble (Tabs. A7,A8). To understand if

the snow cover plays a role in the observed increased sea ice extent, we also analyze its thickness in fall and winter over both the Greenland Sea and the Baffin Bay. The snow cover is notably thicker in both areas under the NSR scenario across all seasons compared to the TSR scenario. Only in fall the difference is minimal, not statistically significant under either scenario compared to the control ensemble. The snow cover plays a dual role in regulating the sea ice extent. In summer, its high albedo reflects a large fraction of the incoming SW radiation, slowing down sea ice loss. In winter, it insulates the sea ice, inhibiting

ice growth (Sturm et al., 2002). Thus, although the thicker snow cover under the NSR scenario during summer would help in maintaining a more extensive sea ice cover, during winter and spring - the sea ice growth seasons - it might lead to the opposite effect. However, BC emissions are unlikely to be the direct cause for the thicker snow cover observed over the Greenland Sea and Baffin Bay, as BC has been shown to reduce precipitation across all atmospheric layers (Sand et al., 2020). Nevertheless, by triggering sea ice loss in the surrounding regions, BC might enhance snowfall over these areas, leading to greater snow

accumulation over the sea ice. A key local effect of sea ice is its ability to reduce surface evaporation from the ocean, limiting the formation of clouds (Vihma et al., 2016). As sea ice loss proceeds, more moisture is supplied to the atmosphere, contributing to enhanced precipitation (Liu et al., 2024). While this could explain the increase in snowfall observed over the Greenland Sea and the Baffin Bay, the complex interplay between sea ice loss and increased precipitation is beyond the scope of this paper, and we therefore cannot draw any definitive conclusions.

**5  Conclusions**

Our idealized experiments reveal that BC deposition from the NSR and TSR affects ASI differently, with impacts occurring throughout the year. The most notable differences between the two routes are seen in terms of net SW radiation at the surface.



Specifically, the NSR impacts the Laptev Sea and Central Arctic in summer, with no significant effects in spring. In contrast, the TSR influences multiple regions in both spring and summer. In spring, we observe increased net SW radiation over the ice in the East Siberian and Laptev seas. In summer, we see increases in the Chukchi and East Siberian seas, and in the Canadian Archipelago. Thus, although the magnitude of these effects may be moderated by interactions with other pollutants not considered in this study, the TSR appears to have a more pronounced and widespread influence on ASI than the NSR. While increases in net SW radiation are not always accompanied by statistically significant changes in sea ice cover or TAS, they remain important. Changes in the sea ice reflectivity could potentially influence long-term trends in ice stability and extent. However, the high computational cost of running CESM2 prevents us from extending our simulations further in this study. Nevertheless, our results underscore the need to consider regional environmental consequences when planning future Arctic navigation strategies.

*Data availability.* The CMIP6 data is freely distributed and available at https://aims2.llnl.gov/search/cmip6/ (Eyring et al., 2016). The CESM2 output for the variables used in our analysis is available at Poltronieri (2024).





**Figure 1. (a)** The red area is the NSR zone, as defined at http://www.nsra.ru/en/ofitsialnaya_informatsiya/granici_smp.html. The blue area represents our approximation of the TSR, encompassing several possible routes crossing the center of the Arctic, depending on the seasonal sea ice distribution. **(b)** MASIE partition of the Arctic (U.S. National Ice Center et al., 2010), adapted from Poltronieri et al. (2024b). The numbers correspond to (1) the Beaufort Sea, (2) the Chukchi Sea, (3) the East Siberian Sea, (4) the Laptev Sea, (5) the Kara Sea, (6) the Barents Sea, (7) the Greenland Sea, (8) the Baffin Bay, (9) the Canadian Archipelago, and (10) the Central Arctic. **(c)** Time series of the injected BC emissions. Due to the structure of the input files in CESM2, the emissions are added monthly every 10 years and interpolated by the model for the missing years in between.







**Figure 2. (a)** Difference in net SW radiation at the surface between the NSR scenario and the control ensemble in spring, averaged over the time period 2041-2050. **(b)** same as **(a)** but for the TSR scenario. **(c)** and **(d)** show the same comparisons as **(a)** and **(b)**, respectively, but for summer.




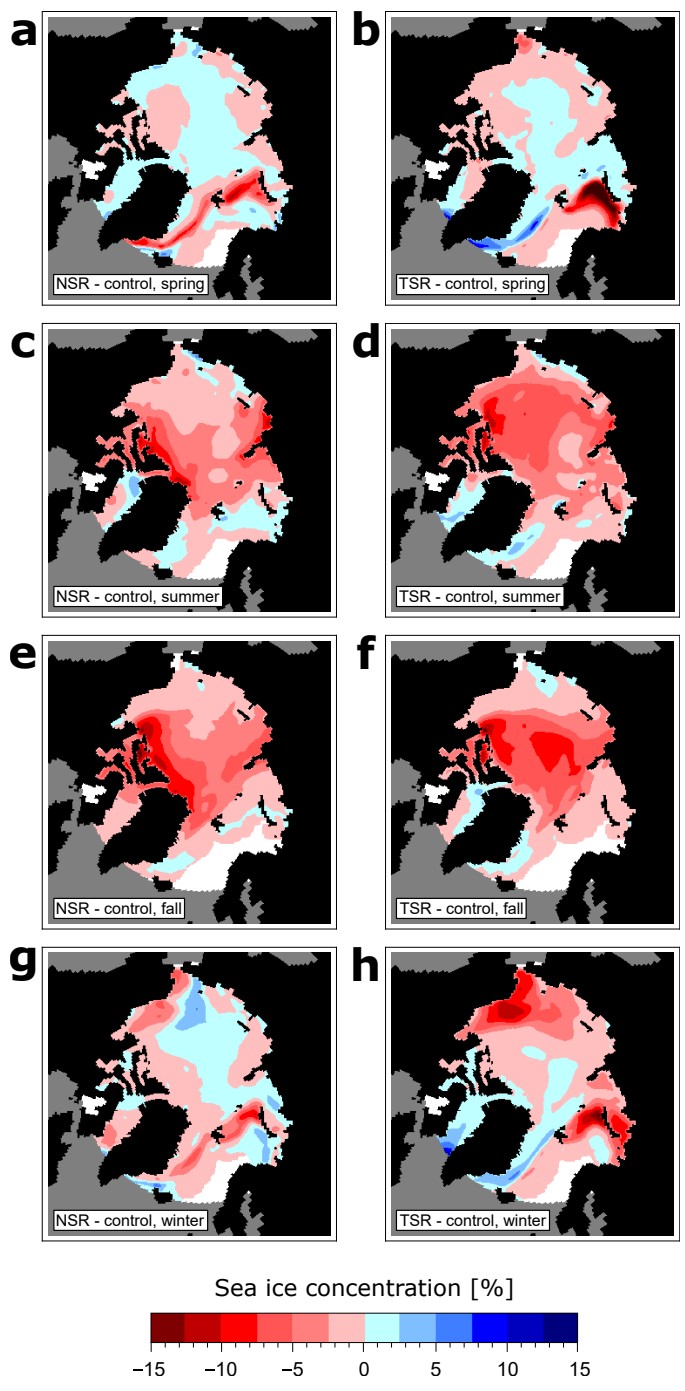

**Figure 3. (a)** Difference in sea ice concentration between the NSR scenario and the control ensemble in spring, averaged over the time period 2041-2050. Red areas denote regions where the NSR scenario shows lower sea ice concentration than the control ensemble. **(b)** same as **(a)** but for the TSR scenario. **(c)**, **(e)**, and **(g)** same as **(a)** but for summer, fall, and winter, respectively. **(d)**, **(f)**, and **(h)** same as **(c)**, **(e)**, and **(g)** but for TSR, respectively.







**Figure 4. (a)** Areas used for grid cell analysis under TSR in spring (see Results section for details). The analysis is performed over all the areas that display enhanced net SW radiation compared to the control ensemble. On the left-hand side, we show the surface composition of the given region. On the right-hand side, we show the difference (TSR minus control ensemble) in net SW radiation at the surface between grid cells with similar sea ice concentration (SIC). The crossed boxes indicate SIC ranges that are not present in the study area. **(b)** same as **(a)**, but for the NSR scenario in summer. **(c)** same as **(a)**, but in summer.



## Appendix A: Period averaged variables

In the following, we list the period-averaged differences in the variables analyzed in our study. The difference is calculated between each navigational scenario - Northern Sea Route (NSR) and Transpolar Sea Route (TSR) - and the control ensemble, averaged over the years 2041-2050. The significance is tested with a two-tailed Student's $t$ test. One (*) and two asterisks (**) indicate high ($p < 0.05$) and very high ($p < 0.01$) statistical significance, respectively.

### A1   Net shortwave radiation

**Table A1. Difference in mean net SW radiation between NSR and control.** Difference in net SW radiation between the NSR scenario and the control ensemble averaged over the time period 2041-2050. The values are given in W/m$^2$.

| MASIE region | Spring | Summer |
|---|---|---|
| Beaufort Sea | 0.84 | 1.55 |
| Chukchi Sea | 0.9 | 0.68 |
| East Siberian Sea | 1.05 | 1.17 |
| Laptev Sea | 1.06 | 4.67* |
| Kara Sea | 0.49 | 2.38 |
| Barents Sea | 0.8 | -0.48 |
| Greenland Sea | 1.41 | -1.51 |
| Baffin Bay | -1.14 | -1.12 |
| Canadian Archipelago | 0.51 | 1.79 |
| Central Arctic | 0.17 | 3.7** |





**Table A2. Difference in mean net SW radiation between TSR and control.** Difference in net SW radiation between the TSR scenario and the control ensemble averaged over the time period 2041-2050. The values are given in W/m$^2$.

| MASIE region | Spring | Summer |
|---|---|---|
| Beaufort Sea | 2.25 | 7.02** |
| Chukchi Sea | 2.97 | 4.62** |
| East Siberian Sea | 1.56** | 4.64** |
| Laptev Sea | 1.03* | 2.82** |
| Kara Sea | -0.71 | 1.84 |
| Barents Sea | 2.5* | -1.11 |
| Greenland Sea | -0.58 | -0.5 |
| Baffin Bay | -0.39 | -2.22 |
| Canadian Archipelago | 0.31 | 4.61* |
| Central Arctic | -0.17 | 2.64* |



## A2   Sea ice area

**Table A3. Difference in mean ASI area between NSR and control.** Difference in ASI area between the NSR scenario and the control ensemble averaged over the time period 2041-2050. The values are given in $10^3$ km$^2$.

| MASIE region | Spring | Summer | Fall | Winter |
|---|---|---|---|---|
| Beaufort Sea | 3.73 | -27.21 | -45.85* | -17.35 |
| Chukchi Sea | 3.83 | -8.31 | -6.98 | 7.08 |
| East Siberian Sea | -1.43 | -5.72 | -17.61 | 9.42 |
| Laptev Sea | -3.15 | -38.67 | -34.45 | -1.72 |
| Kara Sea | -0.92 | -24.6 | -14.07 | 3.89 |
| Barents Sea | -30.62 | -4.12 | -2.16 | -16.17 |
| Greenland Sea | -22.38 | -9.64 | -15.24 | -12.67 |
| Baffin Bay | 32.95 | -1.08 | -16.06 | -2.36 |
| Canadian Archipelago | -0.75 | -32.2** | -40.93** | -1.31 |
| Central Arctic | 0.47 | -125.26* | -166.15 | 5.22 |

**Table A4. Difference in mean ASI area between TSR and control.** Difference in ASI area between the TSR scenario and the control ensemble averaged over the time period 2041-2050. The values are given in $10^3$ km$^2$.

| MASIE region | Spring | Summer | Fall | Winter |
|---|---|---|---|---|
| Beaufort Sea | -5.12 | -63.62** | -45.2* | -50.1* |
| Chukchi Sea | -9.05 | -32.71 | -11.11 | -58.57 |
| East Siberian Sea | -3.98 | -37.13 | -17.58 | -39.06 |
| Laptev Sea | -5.1 | -38.03 | -39.14* | -5.73 |
| Kara Sea | 7.35 | -24.39 | -14.35 | -32.33 |
| Barents Sea | -80.53 | -18.12 | -3.9 | -54.59 |
| Greenland Sea | 32.87 | 0.85 | -5.16 | 13.25 |
| Baffin Bay | 74.53** | 10.71 | 4.45 | 41.89* |
| Canadian Archipelago | -1.63 | -34.2** | -27.99* | -2.44 |
| Central Arctic | 1.85 | -125.47* | -204.31* | -1.34 |





## A3 Cloud cover

**Table A5. Difference in mean vertically integrated cloud cover between NSR and control.** Difference in vertically integrated cloud cover between the NSR scenario and the control ensemble averaged over the time period 2041-2050. The values are given in %.

| MASIE region | Spring | Summer |
|---|---|---|
| Beaufort Sea | -0.49 | 0.33 |
| Chukchi Sea | -0.8 | 0.4 |
| East Siberian Sea | -0.4 | 0.11 |
| Laptev Sea | -0.12 | -0.5 |
| Kara Sea | 0.49 | -0.22 |
| Barents Sea | 0.22 | -0.01 |
| Greenland Sea | 0.14 | 0.34 |
| Baffin Bay | -0.33 | 0.82 |
| Canadian Archipelago | -0.5 | 0.34 |
| Central Arctic | 0.18 | -0.3 |

**Table A6. Difference in mean vertically integrated cloud cover between TSR and control.** Difference in vertically integrated cloud cover between the TSR scenario and the control ensemble averaged over the time period 2041-2050. The values are given in %.

| MASIE region | Spring | Summer |
|---|---|---|
| Beaufort Sea | 0.04 | -0.78 |
| Chukchi Sea | -0.23 | -0.71* |
| East Siberian Sea | 0.45 | -0.82 |
| Laptev Sea | 0.16 | -0.22 |
| Kara Sea | 0.18 | 0.02 |
| Barents Sea | 0.14 | 0.41 |
| Greenland Sea | 0.52 | 0.39 |
| Baffin Bay | -0.02 | 0.69 |
| Canadian Archipelago | 0.47 | -0.39 |
| Central Arctic | 0.72 | 0.09 |





## A4 Snow thickness

**Table A7. Difference in mean snow height over the sea ice between NSR and control.** Difference in snow height over the sea ice between the NSR scenario and the control ensemble averaged over the time period 2041-2050. The values are given in mm.

| MASIE region | Spring | Summer | Fall | Winter |
|---|---|---|---|---|
| Beaufort Sea | 14.61 | -0.05 | -2.69** | 9.49* |
| Chukchi Sea | 28.13* | 1.89 | -0.64 | 21.72** |
| East Siberian Sea | 8.43 | 0.77 | -0.54 | 4.45 |
| Laptev Sea | -2.81 | -1.28 | -1.44* | -1.44 |
| Kara Sea | -0.19 | -0.76 | -0.89* | -2.4 |
| Barents Sea | 24.56 | 1.82 | -0.73* | 7.38 |
| Greenland Sea | 52.** | 5.46** | -3.91 | 16.93** |
| Baffin Bay | 23.39** | 5.63** | 3.47 | 23.32** |
| Canadian Archipelago | 8.98 | -0.86 | -2.08 | 11.88** |
| Central Arctic | 2.9 | -3.71 | -4.58* | -3.41 |

**Table A8. Difference in mean snow height over the sea ice between TSR and control.** Difference in snow height over the sea ice between the TSR scenario and the control ensemble averaged over the time period 2041-2050. The values are given in mm.

| MASIE region | Spring | Summer | Fall | Winter |
|---|---|---|---|---|
| Beaufort Sea | -9.64 | -1.86 | -2.27** | -5.17 |
| Chukchi Sea | -9.97 | -1.09 | -0.96* | -3.45 |
| East Siberian Sea | -0.13 | -0.43 | -0.35 | -2.31 |
| Laptev Sea | -8.24 | -2.73** | -0.89 | -2.05 |
| Kara Sea | -9.86 | -4.75** | -0.99** | -7.64 |
| Barents Sea | -1.88 | -1.98 | -0.92** | -2.74 |
| Greenland Sea | 27.57** | 2.56 | -1.29 | 5.05 |
| Baffin Bay | 12.12** | 3.74** | 2.27 | 9.14** |
| Canadian Archipelago | -1.38 | -2. | -0.06 | 3.48 |
| Central Arctic | 0.99 | -4.95* | -3.4 | -6.34 |





## A5   Surface temperature

**Table A9. Difference in mean TAS between NSR and control.** Difference in TAS between the NSR scenario and the control ensemble averaged over the time period 2041-2050. The values are given in °C.

| MASIE region | Spring | Summer | Fall | Winter |
|---|---|---|---|---|
| Beaufort Sea | -0.2 | 0.06 | 0 | 0.38 |
| Chukchi Sea | -0.3 | -0.14 | -0.41* | -0.1 |
| East Siberian Sea | -0.05 | -0.09 | -0.25 | -0.51 |
| Laptev Sea | 0.28 | 0.29 | 0.17 | -0.19 |
| Kara Sea | 0.03 | 0.27 | -0.24 | -0.17 |
| Barents Sea | -0.11 | -0.03 | -0.47* | -0.31 |
| Greenland Sea | 0.08 | -0.06 | -0.18 | -0.24 |
| Baffin Bay | 0.07 | 0.24* | 0.14 | 0.39 |
| Canadian Archipelago | 0 | 0.41** | 0.37 | 0.24 |
| Central Arctic | 0.31 | -0.22** | 0.28 | 0.16 |

**Table A10. Difference in mean TAS between TSR and control.** Difference in TAS between the TSR scenario and the control ensemble averaged over the time period 2041-2050. The values are given in °C.

| MASIE region | Spring | Summer | Fall | Winter |
|---|---|---|---|---|
| Beaufort Sea | 0.28 | 0.46* | 0.04 | 1.28 |
| Chukchi Sea | 0.25 | 0.29 | -0.13 | 1.36 |
| East Siberian Sea | 0.19 | 0.34 | -0.04 | 0.67 |
| Laptev Sea | 0.06 | 0.33* | 0.12 | 0.2 |
| Kara Sea | -0.07 | 0.05 | 0.07 | 0.39 |
| Barents Sea | 0.18 | 0 | -0.2 | -0.02 |
| Greenland Sea | -0.29 | -0.16 | -0.44** | -0.63 |
| Baffin Bay | -0.29 | 0.25* | -0.63** | -0.27 |
| Canadian Archipelago | 0.16 | 0.65** | 0.02 | 0.46 |
| Central Arctic | 0.24 | -0.22** | 0.31 | 0.63 |





## A6 Melt pond fraction

**Table A11. Difference in mean radiation-effective melt pond fraction between NSR and TSR.** Difference in radiation-effective melt pond fraction between the NSR and the TSR scenarios averaged over the time period 2041-2050. The values are given in %.

| MASIE region | Spring | Summer |
|---|---|---|
| Beaufort Sea | -0.37** | 2.42** |
| Chukchi Sea | -0.27 | 1.79* |
| East Siberian Sea | -0.1 | 0.65 |
| Laptev Sea | 0.01 | -0.64 |
| Kara Sea | -0.01 | 0.02 |
| Barents Sea | -0.1* | 0.41 |
| Greenland Sea | 0. | -5.56** |
| Baffin Bay | 0.06* | -0.09 |
| Canadian Archipelago | -0.06 | 1.3 |
| Central Arctic | -0.01 | -0.87 |



*Author contributions.* A. P. conceived the study, performed the experiments and the analysis. All authors discussed and interpreted the results. A. P. wrote the paper with contributions from the other authors.

*Competing interests.* The authors declare that they have no competing financial interests.

*Acknowledgements.* This is ClimTip contribution #24; the ClimTip project has received funding from the European Union's Horizon Europe
research and innovation programme under grant agreement No. 101137601: Funded by the European Union. Views and opinions expressed are however those of the authors only and do not necessarily reflect those of the European Union or the European Climate, Infrastructure and Environment Executive Agency (CINEA). Neither the European Union nor the granting authority can be held responsible for them. This work was supported by the UiT Aurora Centre Program, UiT The Arctic University of Norway (2020), and the Research Council of Norway (project number 314570). The CESM experiments and parts of the output analysis were performed on resources provided by Sigma2
- the National Infrastructure for High-Performance Computing and Data Storage in Norway under the project nn8008k. We thank Nikolai Figenschau for his help in providing the processed data used in our emission estimates.



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
