# Peer review of "Simulation of future impact of black carbon emissions from the Northern and Transpolar Sea routes on Arctic sea ice"

_EGUsphere, 2025_

## Referee Comment (RC2)

The study uses the global Earth System Model CESM2, to study the potential climate impact of black carbon aerosols emitted from ship traffic through the Northern Sea Route or the Trans Arctic Route.

The study is phrased as quantifying the climate impacts of BC deposited on snow. However, in the simulations the model is forced by emissions in the lowest atmospheric layer. Thus, the impacts are driven by the radiative forcings of BC in the atmosphere as well as BC deposited on the surface. Flanner (2013) estimates that BC residing in the lowest atmospheric layer produces very strong Arctic warming per unit mass and forcing [ $2.8 \pm 0.5$ K/Wm $-$ 2) ], while the annual BC on snow Arctic sensitivity is less at $1.4 \pm 0.7$ K/Wm $-$ 2

My main concern with this study, is that I am not convinced that the results from the perturbation simulations with the model is really statistically significant. It is stated several places in the manuscript that it is, and indeed the statistical t-tests show some apparently significant results. However, this is to be expected as the diagnostics are done for 2 perturbations, 10 regions, 4 seasons, and 5+ climate variables. Then the change in some of these will be significant by chance.

The root cause for this is that the perturbations (added BC from shipping) is very small, of the order of 250 tons/yr. One can make a back-of-the envelope estimate of the radiative forcing based on the calculations from Peters et al (2011) and Fuglestvedt et al (2014). Peters et a. estimate Arctic shipping emissions of BC at 1.0 and 1.8 ktons/yr for 2030 and 2050 respectively. Using these emissions as input to calculate radiative forcing by shifting container ship traffic from the Suez route to the Arctic, Fuglestvedt et al.(2014), estimate a global radiative forcing of the order of 0.1 mWm-2 from BC deposited on snow and ice  (cf their figure 1).  Ødemark et al. (2012) found a radiative forcing of BC from shipping emissions north of 60 deg N (1.6 kt/yr in 2004), to be 0.03 mWm-2 (global) and 0.47 mWm-2 (north of 60N).  Given the expected very low radiative forcing from BC from Arctic Shipping, the expected impact on Surface air temperatures (based on the sensitivities from Flanner, 2013) will be very low (about 0.001 K). To get a robust signal-to-noise ratio in an ESM for such a small perturbation, one would have to run a much larger number of ensembles, or dramatically scale up the perturbation (e.g. as done in Sand et al., 2013).

If the authors decide to re-run their simulations, I would suggest to look into the option to use the results from the CESM2 Large Ensemble Community Project (LENS2) (https://www.cesm.ucar.edu/community-projects/lens2) for the control to save some computer time. In LENS2 100 ensemble members for the high emission scenario SSP3-7.0 is available. However, even 100 ensemble members might be low, and also a high number of perturbation ensemble simulations will have to be done, so possibly also the emissions need to be scaled up.

The authors argue that the scenario SSP5-8.5 is needed as the background to allow for enough melting of ice for the Arctic to become navigate able in 2050. SSP3-7.0 is only slightly less warm, and there has also been arguments that SSP5-8.5 should not be used as the fossil fuel use is simply unrealistically high (Hauzfather and Peters, 2020.)

Specific Comments

Section 2.2.

Emissions of BC from engines are generally at a minimum when the engines are run at optimal design output (e.g. see the black smoke when ships leave harbors at non-optimum engine use). Travelling in

ice, is there a chance that there will be star/stop/accelerations due to ice so that emissions are higher than "normal"? If yes, is this taken into account in the emission inventory?

Is there potential for mitigation of BC-emissions for future shipping? See e.g. Wu et al., 2024.

It would be useful to also get the total annual BC emissions from shipping (e.g. 2050 and 2100) and to put this in context with estimates of BC emissions from other sources within the Arctic. E.g. Sand et al. 2013 use 0.07 Tg/yr (or 70000 tons/yr) for 60-90 N (all sources), while the estimated BC from shipping here is only about 250 tons/yr in 2050 (based in fig 1c).

Section 2.3:

Page 4 line 90: It is stated that the CESM2 model is a "fully-coupled" ESM. Please refrain from using the term "fully-coupled". In my opinion an ESM can not be fully coupled, nature is much too complex for that. I am of course aware that "fully-coupled" in general means that atmosphere and ocean is coupled without flux-adjustments, but I am of the opinion that this terminology is misleading.

Section 3 Results.

The first set of results that are shown and discussed are the changes to SW surface radiation. The model is forced by emissions of BC to the atmosphere, so there are a number of more robust results that should be given. This includes concentration changes of BC in the atmosphere and in the snow (on the sea ice), and importantly the TOA radiative forcing following this. The RF is important because it tells about the expected signal to noise ratio for the coupled ESM simulations.

References:

Hauzfather & Peters. Emissions – the 'business as usual' story is misleading. Nature 577, 618-620 (2020), doi: https://doi.org/10.1038/d41586-020-00177-3

M. Sand, T. K. Berntsen, K. von Salzen, M. G. Flanner, J. Langner and D. G. Victor, Response of Arctic temperature to changes in emissions of short-lived climate forcers, Nature Climate Change 2015, DOI: 10.1038/NCLIMATE2880

Flanner 2013. Arctic climate sensitivity to local black carbonhttps://doi.org/10.1002/jgrd.50176. JGR-Atmospheres.

Fuglestvedt, et al., 2014. Climate Penalty for Shifting Shipping to the Arctic, dx.doi.org/10.1021/es502379d, Environ. Sci. Technol. 2014, 48, 13273–13279

Wu et al., 2024. Enhanced particulate filter with electrostatic charger: Insights for low-resistance and high-efficiency ship-based nanoscale black carbon capture. Process Safety and Environmental Protection, Vol 184. https://doi.org/10.1016/j.psep.2024.02.021